# Alpha-Melanocyte-Stimulating Hormone Maintains Retinal Homeostasis after Ischemia/Reperfusion

**DOI:** 10.3390/biom14050525

**Published:** 2024-04-27

**Authors:** Tat Fong Ng, Jenna Y. Cho, John L. Zhao, John R. Gardiner, Eric S. Wang, Elman Leung, Ziqian Xu, Samantha L. Fineman, Melinda Lituchy, Amy C. Lo, Andrew W. Taylor

**Affiliations:** 1Department of Ophthalmology, Boston University Chobanian & Avedesian School of Medicine, Boston, MA 02118, USA; tatfong@bu.edu (T.F.N.);; 2Department of Ophthalmology, Li Ka Shing Faculty of Medicine, The University of Hong Kong, Pokfulam, Hong Kong SAR, China

**Keywords:** melanocortins, ischemia/reperfusion, retinal degeneration, inflammation, apoptosis

## Abstract

Augmenting the natural melanocortin pathway in mouse eyes with uveitis or diabetes protects the retinas from degeneration. The retinal cells are protected from oxidative and apoptotic signals of death. Therefore, we investigated the effects of a therapeutic application of the melanocortin alpha-melanocyte-stimulating hormone (α-MSH) on an ischemia and reperfusion (I/R) model of retinal degenerative disease. Eyes were subjected to an I/R procedure and were treated with α-MSH. Retinal sections were histopathologically scored. Also, the retinal sections were immunostained for viable ganglion cells, activated Muller cells, microglial cells, and apoptosis. The I/R caused retinal deformation and ganglion cell loss that was significantly reduced in I/R eyes treated with α-MSH. While α-MSH treatment marginally reduced the number of GFAP-positive Muller cells, it significantly suppressed the density of Iba1-positive microglial cells in the I/R retinas. Within one hour after I/R, there was apoptosis in the ganglion cell layer, and by 48 h, there was apoptosis in all layers of the neuroretina. The α-MSH treatment significantly reduced and delayed the onset of apoptosis in the retinas of I/R eyes. The results demonstrate that therapeutically augmenting the melanocortin pathways preserves retinal structure and cell survival in eyes with progressive neuroretinal degenerative disease.

## 1. Introduction

Repeated episodes of ischemia and reperfusion (I/R) in the retina are associated with retinal degenerative diseases, such as diabetic retinopathy (DR) [1,2]. After each episode of I/R, there is a progressive deterioration of the retinal structure and retinal cell loss, leading to impaired vision and potential blindness. The progressive neurodegenerative and microvascular condition in DR affects one-third of the 285 million individuals worldwide and is a leading cause of vision impairment and blindness in middle-aged and elderly adults [3]. It is the most common complication of diabetes mellitus (DM), and the number of individuals affected by DR is expected to increas due to the increasing prevalence and life expectancy of people with type 2 diabetes [4]. Hyperglycemia is the central biomarker of DM. The hyperglycemic levels facilitate the pathogenesis of DR through multiple metabolic pathways, including the polyol, the protein kinase C (PKC), and the hexosamine pathways, with the accumulation of advanced glycation end-products [2].

The changes in metabolic pathways caused by hyperglycemia can induce vessel constriction, pericytes, and endothelial cell apoptosis [2]. The loss of pericytes and endothelial cells changes the architecture of the capillary vessels and causes microaneurysms within the retina, deterioration of the blood–retinal barrier, and retinal ischemia. Over time, retinal ischemia leads to the upregulation of angiogenic factors, such as the vascular endothelial growth factor (VEGF), and the downregulation of anti-angiogenic factors, such as the pigment epithelium-derived factor [2,5,6]. Increased VEGF signaling is strongly associated with increased vascular permeability via the breakdown of tight junctions, resulting in the leakage of blood and exudates from the retinal microvasculature [7]. This diabetic macular edema (DME) is characteristic of the early stage of DR, or non-proliferative diabetic retinopathy (NPDR). Chronic upregulation of VEGF signals for neovascularization within the retina, triggering the later stage of DR, or proliferative diabetic retinopathy (PDR) [2]. However, these new blood vessels are incompetent and prone to microaneurysms, facilitating further DME.

Any sudden return of blood flow to the retina following ischemia may result in inflammatory responses and/or oxidative damage, which may further affect vision [8]. This is known as an ischemia-reperfusion injury (IRI). The IRI results in elevated levels of reactive oxygen species that mediate tissue damage and apoptosis associated with permanent vision impairment and blindness. There is a potential role of immune cells with retinal cell loss in that retinal IRI sequesters CD4+ T cells and macrophages in the retina, facilitating the degeneration of retinal ganglion cells (RGCs) [3,9]. Therapeutic injections of T-cell-blocking antibodies attenuated RGC and retina function in an experimental retinal IRI model. Current treatments for diabetic retinopathy are limited to downregulating VEGF activity because of its associations with macular edema and neovascularization [10]. However, there are no effective treatments available for acute retinal ischemia that can mitigate cell loss and inflammation in IRI. 

Melanocortins are a family of highly conserved neuropeptides and receptors, of which the alpha-melanocyte-stimulating hormone (α-MSH) is the prototype melanocortin neuropeptide [11,12,13]. Melanocortins play a role in pigmentation, metabolism, reproduction, aggression, inflammation, and immunity [11,12,13]. The neuropeptide α-MSH is endogenously produced within the healthy eye by the retinal pigment epithelial cells (RPE) and plays a central role in the normal anti-inflammatory microenvironments of the immune-privileged eye [14,15]. Augmenting the melanocortin pathways with injections of α-MSH suppresses both innate and adaptive immune-mediated inflammation. This suppression involves changing the behavior of immune cells [15]. Immune cells, when treated with α-MSH, after being stimulated by factors that promote inflammation, produce anti-inflammatory cytokines, suppress inflammation, and mediate immune tolerance [15]. Besides the suppression of inflammation, α-MSH rescues cells from apoptosis [14,16,17,18,19,20]. Diabetic mice and rats treated with α-MSH or melanocortin receptor agonists have suppressed pathology and cytokines of diabetic retinopathy, including the retention of viable photoreceptors and ganglion cells [20,21,22,23,24]. In addition, mice with central vein occlusions survive longer with reduced stroke and retinal pathology following α-MSH treatment [25]. The anti-inflammatory and cell-surviving signals of α-MSH may be a viable treatment for acute IRI. Therefore, we analyzed the effects of α-MSH treatment on the pathology, cell survival, and apoptosis in retinas after inducing IRI.

## 2. Materials and Methods

### 2.1. Experimental Ischemia-Reperfusion Injury and α-MSH Treatment 

The mice were 8–10-week-old C56 BL/6J mice purchased from Jackson Laboratories (Bar Harbor, Mount Desert Island, ME, USA). The mice were cared for in the Boston University Animal Science Center (ASC). All experimental uses of the mice followed policies approved by the Boston University Institutional Animal Care and Use Committee (IACUC) and the ARVO Statement for the Use and Care of Animals in Ophthalmic and Vision Research. Mice were anesthetized with an intraperitoneal injection of ketamine (Zoetis, Kalamazoo, MI, USA) and xylazine (Covetrus, Dublin, OH, USA). A drop of 1% tropicamide (Akorn, Lake Forest, IL, USA) solution was placed in the eye to dilate the pupil, followed by a drop of 0.5% topical anesthetic proparacaine (Alcon, Fort Worth, TX, USA). The anterior chamber of the mouse eye was cannulated using a 30-gauge needle attached to a bag of phyiological saline (Fisher Scientific, Hampton, NH, USA) elevated to 1.2 m, or approximately 90 mmHg, for one hour [26]. The saline bag was lowered to the same level as the mouse for 5 min to normalize the intraocular pressure (IOP) before the removal of the needle from the anterior chamber. The control sham I/R group included mice that received similar cannulation into the anterior chamber but without IOP elevation or treatment.

The mice receiving I/R injury received two 1 µL anterior chamber injections of α-MSH (1 μg/mL) (Bachem, Torrance, CA, USA) one day and five days after I/R. The injections were performed by using a 30-gauge needle attached to a 10 μL Hamilton 700 series micro-syringe (Hamilton Co., Reno, NV, USA) that followed the I/R cannula tract through the cornea. The untreated I/R mice were injected with 1 µL of PBS carrier. Seven days after I/R, the eyes were collected. The enucleated eyes were fixed in 4% paraformaldehyde (Fisher Scientific) for 48 h, embedded in paraffin, and sliced into 5 mm sections. The sections were stained with H&E or antibodies for TUNEL. 

### 2.2. Immunohistochemistry

After de-paraffin, the sections were rehydrated, and the antigens were unmasked using an antigen-unmasking solution (Vector Laboratories, Burlingame, CA, USA). The slides were placed in a BioCare Medical Antigen Decloaker (Pacheco, CA, USA) for 15 min at 95 °C; after cooling to 60 °C, they were transferred to PBS solution for immunostaining. The sections were stained with mouse anti-GFAP IgG Cy3-tagged (Sigma-Aldrich, St. Louis, MO, USA, Cat #: C9205); rabbit anti-Iba1 IgG (Wako, Richmond, VA, USA, Cat #: 019-19741); or rabbit anti-glutamine synthetase IgG (Abcam, Waltham, MA, USA, Cat #: Ab73593), with goat anti-rabbit Cy2 (Jackson ImmunoResearch, West Grove, PA, USA, Cat #: 111-225-144) secondary antibody. For glial cell staining, the sections were stained with rabbit anti-Brn3a IgG (Abcam, Cat #Ab245230) with an ImmPRESS horse anti-rabbit IgG Polymer Alkaline Phosphatase kit (Vector Laboratories, Cat# MP-5401). The H&E-stained slides and IHC-stained slides were viewed under a 200× Olympus CX33 Biological Microscope and imaged using a QColor camera system and software v3.1.3.10 (Olympus, Tokyo, Japan). The hematoxylin-stained nuclei or Brn3a-stained nuclei in the ganglion cell layer were counted. ImageJ analysis software v1.54 measured the length of the retinal ganglion cell layer from the optic disc. The nuclei density was plotted in accordance with the treatment group and analyzed. For each mouse eye collected, 3 sections were taken and counted for nuclei. 

### 2.3. Histopathological Scoring 

Images of H&E-stained sections were taken at 100× magnification. We adopted the histopathological scoring for experimental autoimmune uveitis [27] and modified it to I/R injury (Table 1). There were 8 parameters used to identify the injury conditions of the retina: RPE damage; retinal folds; cell loss in GCL, INL, and ONL; photoreceptor loss; and hemorrhage in the vitreous and the subretinal space (See Appendix A). The following table shows the scores of each parameter.

### 2.4. Assay for Apoptosis 

The retinas of the eyes 7 days after I/R were assessed for apoptosis by TUNEL staining. In addition, other groups of mice received a single α-MSH (1 ng) injection into the anterior chamber immediately after the removal of the cannula used to induce I/R, and their eyes were collected 1, 4, 24, and 48 h after I/R and α-MSH treatment. The retinal sections were created, and the staining for TUNEL was performed using a MilliPore ApopTag Red Apoptosis Detection Kit (Millipore, Burlington, MA, USA). After rehydration, the sections were digested with proteinase K, and the sections were treated with terminal deoxynucleotidyl transferase (TdT) enzyme for 2 h at 37 °C. Enzyme activity was stopped, and the sections were then stained with rhodamine-conjugated anti-digoxigenin antibody with a DAPI counterstain. Digital images of the stained sections were taken at 40× and 10× magnifications under fluorescence and analyzed with ImageJ (NIH). The number of TUNEL-positive (apoptotic) cells was counted, and the area of each retina was measured for each section (see Appendix A). The results are presented as the number of TUNEL-positive cells per mm^2^ of the retina.

### 2.5. Statistical Analysis

Statistical significance was determined using an ordinary one-way ANOVA with a Šídák post-analysis multiple comparisons test. A nonparametric Mann–Whitney *t*-test determined the statistically significant differences in histology scores; for the multiple comparisons test, a Benjamini, Krieger, and Yekutieli false discovery method was used. A 2-way ANOVA analyzed the statistically significant differences in the TUNEL assays over time. The day-7 TUNEL-staining results were assayed by an unpaired *t*-test. Statistical analysis was performed using GraphPad Prism 10.2, with *p*-values ≤ 0.05 considered significant and q ≤ 5% considered a discovered difference. All measurements were performed on 5–8 eyes per group. 

## 3. Results

### 3.1. The Effects of α-MSH Treatment on Retinal Structure and Ganglion Cell Survival following I/R 

To assess the effects of α-MSH treatment on the pathology of retinas with IRI, the eyes were collected, sectioned, H&E-stained, and imaged for histological examination (Figure 1). The stained histological images showed that there was damage and loss of cells in the retinas seven days after I/R when compared to retinas from the control sham eyes (Figure 1A). The retinas of the α-MSH-treated IRI eyes retained normal layers of the retina and, in comparison with the untreated IRI eyes, appeared to have less pathology and cell loss (Figure 1A). The retinal images of each eye were histologically scored (Table 1), and the total histological score for each eye was calculated (Figure 1B). The histopathological scores of the retinas from α-MSH-treated IRI eyes were significantly (*p* ≤ 0.005) lower than retinas from IRI eyes (Figure 1B). The reduction in individual histopathological criteria scores was seen in most layers of the retinas from α-MSH-treated eyes, compared with the retinas from untreated IRI eyes (Figure 1C). Although hemorrhaging was seen in some retinas, there was none seen in the subretinas of α-MSH-treated IRI eyes.

Since ganglion cell drop is an early and major consequence of elevated intraocular pressure and I/R [28,29], there was a significant decrease in the density of nucleated cells in the retinas of untreated IRI eyes, compared to the density of nuclei in the control sham retinas (Figure 1D). There was no significant decrease in the ganglion cell layer nuclei density in the retinas of α-MSH-treated IRI eyes. To see if these nucleated cells were viable ganglion cells, the retinal sections were stained with the ganglion cell marker brn3a [30]. The retinas from both the sham eyes and α-MSH-treated IRI eyes expressed brn3a, with a similar pattern of mostly ganglion cells interspersed with a few other brn3a^−^ cells of the ganglion cell layer (Figure 2A). The ganglion cell layer of the untreated IRI eyes had less stained cells, with marked areas of cell loss. There was no significant difference in the number of brn3a^+^ cells along the retinal section between the retinas from sham eyes and α-MSH-treated IRI eyes; however, untreated IRI had a significant decline in the number of brn3a^+^ cells (Figure 2B). These results demonstrated that α-MSH treatment protects the retina from pathology and promotes the survival of ganglion cells from IRI.

### 3.2. The Effects of α-MSH Treatment on Muller and Microglial Cell Activation 

To assay for an injury response within the retinas of IRI eyes, the retinal sections were stained for GFAP and glutamine synthase (GS) for Muller cells (Figure 3) [31,32]. There was no upregulation of GFAP expression in the retina of sham control eyes. There was a great increase in GFAP expression in the retinas from both untreated and α-MSH-treated IRI eyes (Figure 3A). While the number of GFAP^+^ Muller cells was lower in the retinas of α-MSH-treated IRI eyes, it was still significantly higher than in the retinas of sham control eyes (Figure 3B). 

When the retinas from IRI eyes were stained for Iba1^+^ microglial cells [33], there was a marked presence of the microglial cells in the outer layers of the retina in comparison with the sham control (Figure 4A). Also, there was a significant increase in the density of Iba1^+^ microglial cells in the retinas of untreated IRI eyes (Figure 4B). Following treatment with α-MSH, the Iba1^+^ microglial cells were still seen in the outer layers of the retina (Figure 4A) and had a density that was not statistically different from the retinas of the control sham eyes but also not statistically different from the retinas of untreated IRI eyes (Figure 4B). The results showed that following I/R, there was a strong activation of Muller cells, with migration and increased numbers of Iba1^+^ cells in the retina. Treatment with α-MSH had little effect on Muller cell activation and the migration of Iba1^+^ cells in the IRI retinas; however, α-MSH treatment reduced the number of Iba^+^ cells in the IRI retina, suggesting suppression or inhibition of macrophage infiltration and subsequent inflammatory responses. 

### 3.3. The Effects of α-MSH Treatment on Retinal Apoptosis following I/R

Apoptosis is considered the major reason for cell loss in the IRI retina [34]. There were two groups to see if α-MSH had an effect on apoptosis in the IRI retina. One group of mice went through the I/R procedure and α-MSH treatment, as shown above, and the eyes were collected for the TUNEL assay 7 days after I/R (7-day group). The second group of mice was treated with α-MSH in the anterior chamber immediately after the I/R procedure, and the eyes were collected for TUNEL assay at 1, 4, 24, and 48 h. The TUNEL-stained retinas of untreated IRI eyes showed that apoptosis (TUNEL^+^ cells) occurred within hours after I/R (Figure 5A). TUNEL-staining progressed from the ganglion cell layer at 1 h after I/R to the inner nuclear layer at 24 h and the outer nuclear layer within 48 h after I/R (Figure 5A). In comparison to the immediate time after I/R, the number of TUNEL^+^ cells was diminished in the retinas 7 days after I/R, with the retinas from α-MSH-treated IRI eyes having a significantly (*p* ≤ 0.05) lower number than the untreated group (Figure 5B). The average cumulative numbers of TUNEL^+^ cells measured from eyes 1, 4, 24, and 48 h after I/R showed that there was a significant overall reduction in TUNEL staining in the retinas from α-MSH-treated IRI eyes (Figure 5C). This demonstrates that α-MSH suppressed the signals of apoptosis in the retina following I/R, which helped preserve renal structure and cell viability.

## 4. Discussion

The melanocortin pathways play an important role in ocular immunobiology, including suppression of inflammation, maintenance of ocular immune privilege, and immune tolerance to ocular antigens [14,15,35]. The melanocortin pathways are active, due to the constant presence of the neuropeptide α-MSH within the normal eye [36,37]. An important source of α-MSH is the RPE, which influences the activity of immune cells within the retina. In addition, α-MSH may well play an important role in promoting retinal cell development and viability, along with inhibiting signals of apoptosis [14,38,39,40,41,42]. Therefore, augmenting the melanocortin pathways with α-MSH may provide a strong signal for the survival and preservation of retinal tissues under inflammatory or stressful conditions.

Treatment of diabetic rodent models with α-MSH suppresses the retinopathy and also suppresses the extent of retinal damage caused by central vein occlusion in diabetic stroke mice models [21,24,25,43]. The actions of α-MSH are not restricted to diabetic conditions and are mostly on the cellular response to injury and inflammation [14,15,17]. To see the effects of α-MSH treatment on retinal damage, we examined the retinas in eyes treated with α-MSH following I/R, which is a model of human retinal pathologies of glaucoma, diabetic retinopathy, and retinal vascular occlusions [8]. Our results demonstrated that there was a significant decrease in the histopathology of the I/R retinas following α-MSH treatment, with a noted decrease in the loss of cells in the ganglion cell layer. This was associated with both the continued survival of ganglion cells and the suppression of apoptosis throughout the retina, as well as with the slowed onset of apoptosis. While α-MSH treatment reduced the number of macrophages/microglia cells in the retina following I/R, it did not change the number of GFAP-positive Müller cells. These results indicate that augmenting the melanocortin pathways with α-MSH treatment effectively promotes retinal cell survival and retinal structure following IRI, which may also provide protection for retinal function.

The pathology of several retinal diseases can be lessened by α-MSH treatment. In experimental autoimmune uveitis, α-MSH treatment suppresses the intraocular inflammation, restores RPE anti-inflammatory activity, and promotes the induction of immune tolerance to retinal antigens, along with preserving the structure of the retina [14,35,44]. The neuropeptide suppresses the migration and infiltration of polymorphonuclear leukocytes into the anterior chamber of eyes with endotoxin-induced uveitis [45,46]. In diabetic mice, α-MSH treatment suppresses the pathology of diabetic retinopathy by promoting the survival of both ganglion cells and photoreceptors, with the retention of the blood–ocular barrier and the inhibition of the induction of neovascularization [21,24,25,43]. The loss of photoreceptors in RCS rat retinas is retained with α-MSH treatment [38]. At the cellular level, α-MSH treatment makes macrophages, and possibly microglial cells, inhibit inflammation and make other immune cells anti-inflammatory [14,47,48,49]. Also, α-MSH suppresses neutrophil activation and migration [50,51,52]. Stimulated ARPE-19 cells treated with α-MSH are suppressed in the production of monocyte chemotactic protein 1 and induce anti-oxidative enzyme activity in primary RPE exposed to high glucose [17,53,54]. Treatment of human corneal buttons for transplantation retains epithelial cells with less cell death [16]. The neuropeptide prevents apoptosis in cultured macrophages and RPE cells [17,55]. In addition, the effects of augmenting the melanocortin pathways may be associated with which melanocortin receptor (MCr) is stimulated. While stimulating MC1r and MC3r effectively suppresses inflammation, MC5r mediates immune tolerance and the induction of suppressor macrophages [14,15]. Targeting MC1r and MC5r in the retina may be important in preventing damage and cell loss in the retina in uveitis, diabetes, and IRI [24,56,57]. Simulating MC4r can promote neurite growth by injured ganglion cells [41]. How each of these melanocortin receptors affects different cellular activities and whether targeting specific melanocortin receptors is effective in preventing the pathology of IRI is yet to be seen. Collectively, these demonstrate the importance of the melanocortin pathways in regulating inflammation and cell survival. 

Our results showed that α-MSH treatment mediated similar protective features, as seen in the other treated diseases [25,58]. We demonstrated that there was protection of the retinal structure, diminished ganglion cell loss, and apoptosis. In contrast, there was still significant Muller cell activation, and an increase in macrophage/microglial cell numbers was noted. Based on the cellular effects of α-MSH, it is very likely that the macrophage/microglial cells we observed were mediating anti-inflammatory activity and suppressing oxidative activity [14,15,35,47,59,60], which needs to be tested. The activated Müller cells could also be important in releasing molecules to further protect the retina from the damaging effect of the I/R procedure [32,61,62]. Since both the macrophages/microglial cells and Muller cells can have both a positive or negative effect on the retinal pathology, the actions will have to be further assayed [63,64,65,66]. Since the retinas after I/R and α-MSH treatment are not significantly different structurally and cellularly from the retinas of normal eyes, it is reasonable to expect that α-MSH induces retinal protective activity in the macrophage/microglial cells and Müller cells. Moreover, the TUNEL staining results suggest that treatment with α-MSH may provide greater protection by inhibiting or diminishing the early-phase signaling of apoptosis.

It should be noted that this work only tested one concentration of α-MSH in the experiments. Higher concentrations may affect the activation of the Muller cells and decrease the number of macrophages. However, the concentration injected intracamerally was higher than the normal concentration of α-MSH in the eye. This means that the melanocortin pathways were well stimulated. Another limitation is that retinal folding makes it unclear how much neuronal loss occurred in the INL and ONL. While we did not measure photoreceptor signaling, it could be another way to determine the effectiveness of α-MSH on retinal neural cell survival. Previous studies have revealed that α-MSH-treated diabetic and I/R diabetic eyes have protected retinal functions measured by ERG [21,25]. 

The results presented further help to define the role of α-MSH, which is constantly present in the normal eye, in maintaining healthy retinas by controlling immune responses and inflammation, safeguarding the blood–retinal barrier, preventing neovascularization, and promoting retinal cell survival. In addition, the results showed that α-MSH treatment alleviated the damage of retinal IRI by addressing the current absence of effective therapeutic interventions for retinal IRI.

## Figures and Tables

**Figure 1 biomolecules-14-00525-f001:**
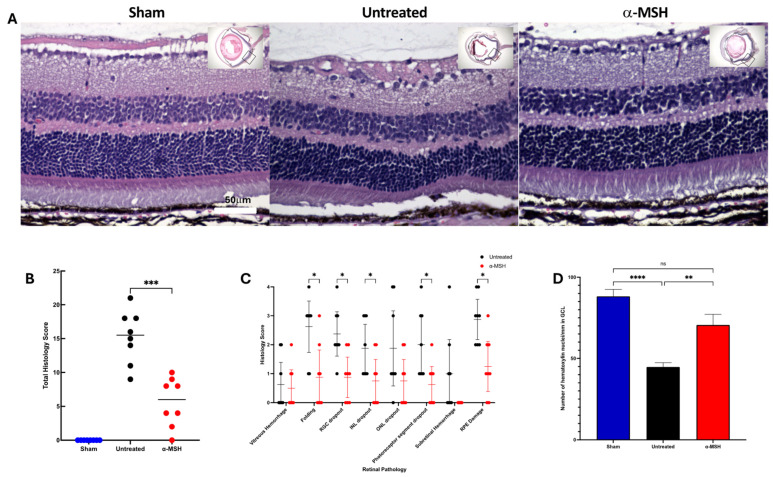
Effects of α-MSH treatment on the histopathology of I/R. Seven days after I/R and α-MSH treatment, the eyes were collected and sectioned. Following H&E staining, images of whole retinal sections were analyzed. (**A**) Representative images of retinas for each group of sham, untreated, and α-MSH-treated I/R-eyes. The images are presented at 40× the central retina, with a whole eye insert showing where the image was taken. (**B**) The total histological score of each mouse in each group is presented, and the mean score is indicated. Retinas from I/R eyes treated with α-MSH had significantly (*p* ≤ 0.005) lower total histopathology scores, compared to untreated I/R eyes. (**C**) The mean ± 95% confidence intervals for each of the individual histopathological criteria scores of the retinas from the untreated and α-MSH-treated I/R eyes are presented. Nonparametric discoveries (*) of less than 5% differences were found between α-MSH-treated and untreated I/R eyes for most of the scored criteria. (**D**) The hemotoxylin nuclei along the ganglion cell layer were counted and presented as nuclei per mm of retinal length mean ± SEM. There was a significant (*p* ≤ 0.001) loss of nuclei in the ganglion cell layer of untreated I/R eyes, with the retention of nuclei in the α-MSH-treated I/R eyes. The ns indicates no statistically significant differences; ** *p* ≤ 0.01, *** *p* ≤ 0.005, and **** *p* ≤ 0.001 of the eight retinal sections per group assayed.

**Figure 2 biomolecules-14-00525-f002:**
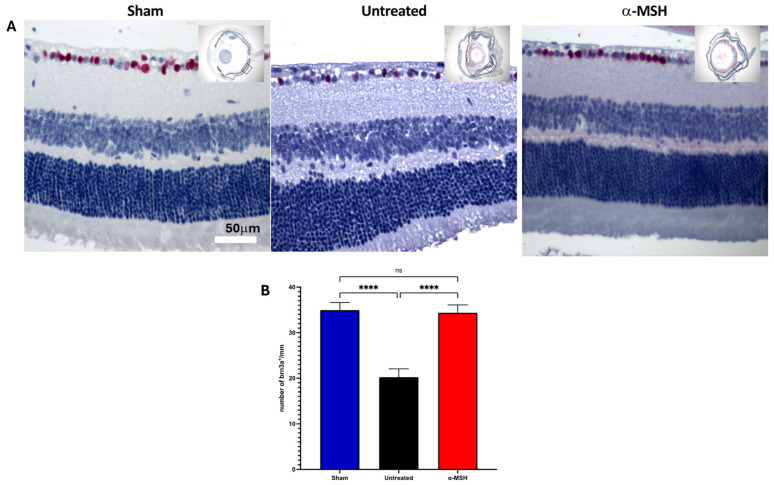
Effects of α-MSH treatment on ganglion cell viability following I/R. The retinal sections were stained for the ganglion cell marker brn3a. (**A**) Representative images of retinas with brn3a-stained nuclei for each treatment group of sham, untreated, and α-MSH-treated I/R-eyes, with 5 eyes per group. The images are presented at 40× the central retina, with a whole eye insert showing where the image was taken. (**B**) The brn3a-stained nuclei along the ganglion cell layer were counted and presented as nuclei per mm of retinal length (mean ± SEM). There was a significant (*p* ≤ 0.001) loss of nuclei in the ganglion cell layer of untreated I/R eyes, with the retention of nuclei in the α-MSH-treated I/R eyes, compared to shame retinas. ns = not statistically significant; **** *p* ≤ 0.001.

**Figure 3 biomolecules-14-00525-f003:**
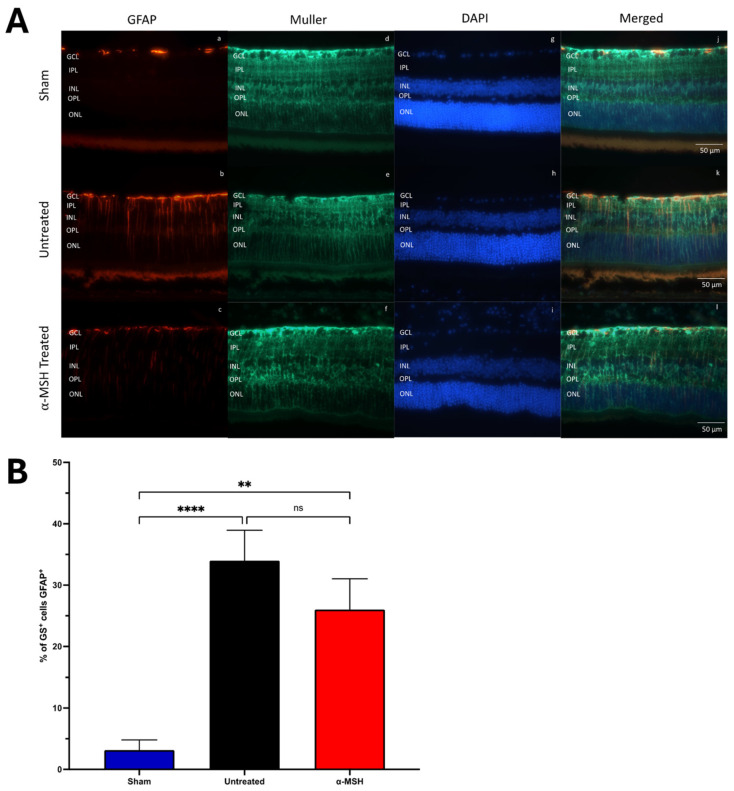
Effects of α-MSH treatment on Müller cell GFAP expression following I/R. The sections were stained for Müller cells with glutamine synthetase (GS) and glial fibrillary acidic protein (GFAP), with a DAPI counterstain. The images were assayed for the number of Muller cells (GS^+^ cells) and GFAP^+^ cells. (**A**) Representative images of stained retinal tissues showing each treatment group’s red-stained GFAP and green-stained Müller cells (**a**–**l**). (**B**) The percent (mean ± SEM) of GS^+^ cells that were GFAP^+^ is presented for each treatment group. There was a significant increase in GFAP-positive Müller cells in both untreated (*p* ≤ 0.001) and α-MSH-treated (*p* ≤ 0.01) I/R eyes. For the 5 retinas assayed per group, ns = not statistically significant, ** *p* ≤ 0.01, and **** *p* ≤ 0.001.

**Figure 4 biomolecules-14-00525-f004:**
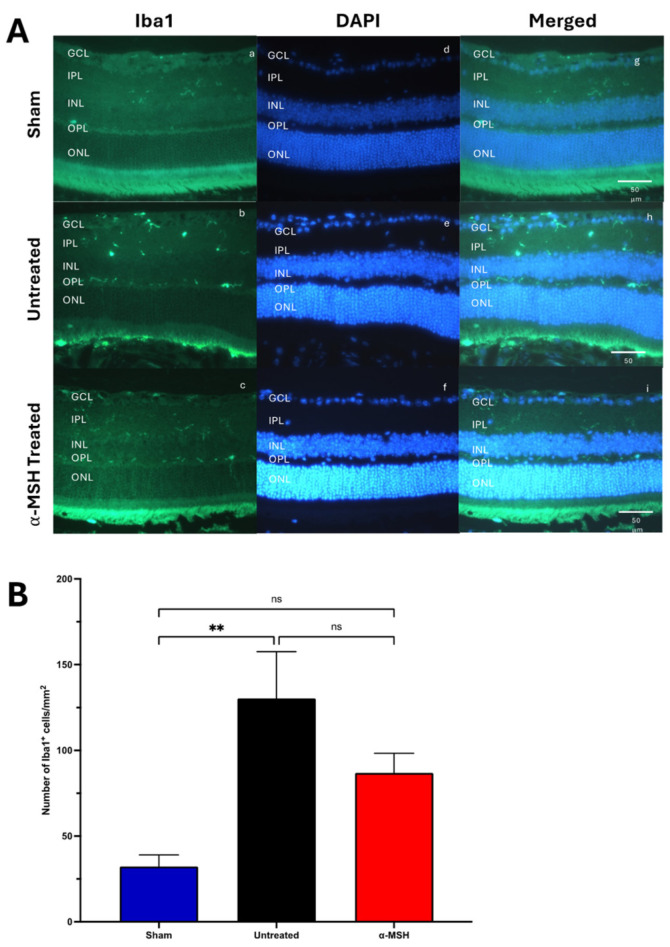
Effects of α-MSH treatment on retina microphage/microglial cells following I/R. The sections were stained for microglial cells with antibodies to Iba1 and counterstained with DAPI. The images were assayed for the number of microglial (Iba1^+^) cells over the area (mm^2^) of the retina. (**A**) Representative images of stained retinal tissues showing the green-stained iba1 cells for each group. The most noted change was the location of the Iba1^+^ cells in the OPL of the untreated I/R eyes versus the α-MSH-treated eyes (**a**–**i**). (**B**) The number (mean ± SEM) of Iba1^+^ cells per area of the retina was measured. There was a significant increase in iba1^+^ cells in untreated (*p* ≤ 0.01) but not in α-MSH-treated I/R eyes. For the 5 retinas assayed per group, ns = not significantly different, and ** *p* ≤ 0.01.

**Figure 5 biomolecules-14-00525-f005:**
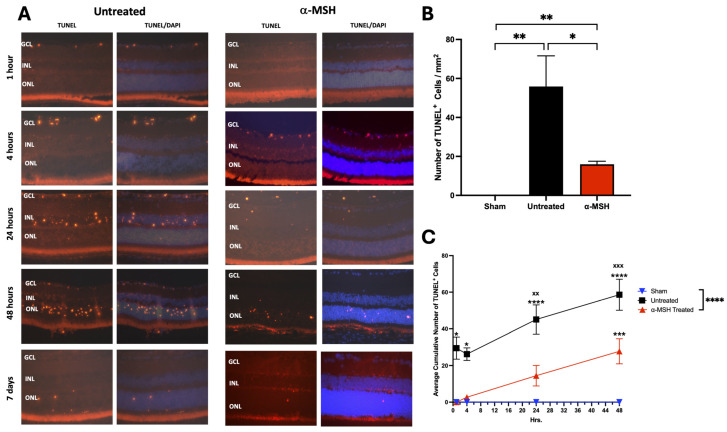
Effects of α-MSH treatment on apoptosis in retinas following I/R. The retinal sections were TUNEL-stained and counterstained with DAPI. (**A**) Representative images of stained retinal tissues showing each group’s red-stained TUNEL and the blue DAPI counterstain. The I/R eyes were treated immediately with an injection of α-MSH after the I/R, and the eyes were collected, sectioned, and stained at 1, 4, 24, and 48 h. The seven-day α-MSH-treated images are retinal sections from eyes treated with α-MSH, as in the other figures. In the untreated group, there was a noticeable progression of TUNEL-positive cells, starting with the ganglion cell layer migrating to the outer nuclear layer. By day 7, most of the TUNEL staining was diminished in all the groups. Retinas from 5 eyes were assayed per treatment group and sham (normal) eyes. (**B**) At 7 days after the I/R, there was no significantly higher density (cells/mm^2^) of TUNEL^+^ cells across the whole retinal section of α-MSH-treated I/R eyes, compared to the sham eyes. * *p* ≤ 0.05, and ** *p* ≤ 0.01. (**C**) The number of TUNEL^+^ cells in I/R eyes immediately treated with α-MSH was counted over 1, 4, 24, and 48 h. For each group, the accumulated number of TUNEL^+^ cells was calculated and presented as the mean ± SEM of the accumulated number of TUNEL^+^ cells over time. Significant accumulation of TUNEL^+^ cells occurred within 4 h of the I/R and continued at all times afterwards for retinas in untreated I/R eyes, compared to sham eyes. The α-MSH-treated I/R eyes showed significance only at 48 h, compared to sham eyes. At 24 to 48 h, there was a statistical difference between the untreated and α-MSH-treated I/R eyes, and the area under the curve was highly significantly (**** *p* ≤ 0.0001) different. Differences with sham eyes were * *p* ≤ 0.05, *** *p* ≤ 0.005, and **** *p* ≤ 0.001, and differences between untreated and α-MSH-treated I/R eyes were ^xx^
*p* ≤ 0.01 and ^xxx^
*p* ≤ 0.005.

**Table 1 biomolecules-14-00525-t001:** Histopathological Scoring.

Parameters		Scores
**1**	** RPE Damage **	0: None
		1: Trace/Mild
		2: Moderate
		3: Severe
		4: Very severe
**2**	** Retinal folds **	0: None
		1: One to two folds
		2: Three to four folds
		3: Five to six folds
		4: Seven or more folds
	** Cells Loss **	
**3**	•Ganglion Cell Layer	0: <1%
**4**	•Inner Nuclear Layer	1: 1–24%
**5**	•Outer Nuclear Layer	2: 25–49%
**6**	•Photoreceptor	3: 50–74%
		4: 75–100%
	** Hemorrhage **	
**7**	•Vitreous	0: None
**8**	•Subretinal	1: Trace
		2: Mild (<5% area)
		3: Moderate (6–10% area)
		4: Severe (>10% area)

## Data Availability

The authors will make the raw data supporting this article’s conclusions available upon request.

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
