# Peer review of "Alpha-Melanocyte-Stimulating Hormone Maintains Retinal Homeostasis after Ischemia/Reperfusion"

_biomolecules, 2024, doi:10.3390/biom14050525_

Round 1
Reviewer 1 Report
Comments and Suggestions for Authors
Only two minor concerns.
(1) For Fig 4 caption title, the authors could refer to "microglia/macrophages" as they did throughout the manuscript.
(2) In Discussion lines 426-427 "Since both the macrophages/microglial cells and Muller cells can have both a positive or negative effect on the retinal pathology.." Perhaps a citation for protective microglia is warranted. These are 2 important ones : PMID: 38289348, PMID: 30850344
Author Response
We thank the reviewer for taking the time to review the manuscript.
(1) For Fig 4 caption title, the authors could refer to "microglia/macrophages" as they did throughout the manuscript.
Corrected
(2) In Discussion lines 426-427 "Since both the macrophages/microglial cells and Muller cells can have both a positive or negative effect on the retinal pathology.." Perhaps a citation for protective microglia is warranted. These are 2 important ones : PMID: 38289348, PMID: 30850344
These references were added to the two we already have for this sentence. Thank you for bringing these two important papers to our attention.
Reviewer 2 Report
Comments and Suggestions for Authors
Ng and collaborators studied the protective effects of the melanocortin alpha-melanocyte-stimulating hormone in mouse eyes afflicted with an ischemia and reperfusion model for retinal degeneration. The authors found that a-MSH prevents retinal deformation, ganglion cell loss, and delays the onset of apoptosis in affected retinas. Although a-MSH does not prevent Muller activation, it also reduces the density of microglial cells at 7 days after IR. The materials and methods are mostly described in sufficient detail, however, although the statistical analyses seem appropriate, the sample size for the different experiments is not clear. Overall, the study suggests that melanocortin may preserve cell survival in eyes with progressive degeneration caused by I/R. Although the authors have shown the efficacy of the melanocortin alpha-melanocyte-stimulating hormone in preventing neuronal loss (Brn3a and TUNEL assays), I think that glial responses would make more sense to study at earlier stages, and TUNEL signal at 24-48h do not seem to correlate with the thickness of the different retinal layers at 7d. English is mostly well-written. Below are some points to consider and a few suggestions to further improve the manuscript.
1. I/R model is characterized by quick damage and degeneration. As the authors have demonstrated most of the degenerative events (TUNEL) occur within 2 days (48 hours). Then, analyzing glial reactivity (Muller cells, microglia cells, and astrocytes) would more sense at that time instead of at 7 days post I/R when degeneration is minimal. I understand that evaluation of Brn3a at 7 days confirms RGC survival, but if reactivity of glial cells is prevented by a-MSH it would be more relevant at early stages, especially when a-MSH presents a significant protection within 2 days (TUNEL assays).
2. Line 123. Regarding the administration of a-MSH. Could the authors explain why the a-MSH was injected into the anterior chamber instead of Intravitreous? Would intravitreous administration improve the a-MSH efficacy?
3. As showed by the TUNEL results, single administration delayed the apoptotic events and protected neurons. Then, why for evaluating the effects of a-MSH on the RGCs survival, Muller cells, microglia cells, and astrocytes reactivity at 7 days, the a-MSH was administrated twice (1 and 5 days after I/R). Have the authors evaluated if a single administration was also protective?
4. Lines 137-140. To facilitate reproducibility, please add the concentration of use and catalog number for the antibodies.
5. Lines 137-140. Material and methods. I understood that Brn3a was identified by an anti-rabbit IgG ImmPRESS-AP. But I think the following sentence is confusing: “Rabbit anti-Iba1 IgG (Wako, Richmond, VA); Rabbit anti-Glutamine Synthetase IgG (Abcam, Waltham, MA) or Rabbit anti-Brn3a IgG (Abcam) with Goat anti-rabbit Cy2 (Jackson ImmunoResearch, West Grove, PA)”, and then: “For Brn3a staining, the secondary antibody was horse anti-rabbit IgG ImmPRESS-AP reagent”. Did the authors mean that Goat anti-rabbit Cy2 was used for Iba1 and Glutamine Synthetase, while the horse anti-rabbit IgG ImmPRESS-AP was for Brn3a? Can the authors clarify that?
6. Number of animals used in the experiment and sample size some experiments are missing. E.g., for quantifications of H&E nuclei in GCL (Fig 1D), Brn3a (Fig 2), and TUNEL (Fig. 5). For other experiments the author expressed that the sample size was n=5 (Fig. 3 legend, line 283; and Fig. 4 legend, line 325), but does it mean n=5/group?
7. Hemorrhages for the histology score. Can the authors explain how to determine if hemorrhages are a consequence of the elevated I/R damage or the surgical procedure? And can the authors include images of hemorrhages for readers' interpretation in Fig. 1? And have the authors investigated the hemorrhages ‘in vivo’ by examining the eye fundus with the OCT?
8. How to determine the damage in the RPE for the histology score. E.g., the RPE in the detailed image for the “untreated group” (Fig. 1A) is damaged, but it seems to be due to the tissue processing/sectioning.
9. Fluorescent staining in paraffin sections. Paraffin sections may not be the best for fluorescence. Cryosections would be a proper tissue processing for immunofluorescence. E.g., staining for Iba1 for the sham group (Fig. 4A) seems a bit weak and MCs seem absent in most of these layers. However, microglial cells are known to be resident microglia extended in at least 3 layers (NFL/GCL, IPL, OPL) in control conditions.
10. Can the authors explain how to determine a positive microglial cell? In Fig. 4 most of the microglial cells are seen partially (the soma, or only some of their processes) ...
11. Line 323-324. Please double-check the following sentence because I do not observe microglial cells located in the ONL in the untreated I/R retinas: “The most noted change was the location of the Iba1+ cells in the ONL of the untreated I/R eyes versus the a-MSH-treated eyes”.
12. Line 293. The suppression of the inflammatory response has not been proved in the present manuscript. Reduction of microglial cells does not prove reduced inflammatory response, because reduced microglial cells could be a consequence of the reduced RGC loss. To prove that additional experiments show a decrease in inflammatory factors, cytokines, etc.
13. Apoptosis (TUNEL) and anatomical correlation. I would expect a significant thinning of the INL and ONL at 7 days if they present a high number of apoptotic cells at 24-48 hours. However, the thickness of INL and ONL seems comparable to the sham/control group. How can those results be conciliated? Did the authors evaluate neuronal and photoreceptor loss in the INL and ONL respectively by quantifying the number of DAPI-positive rows?
14. Fig. 5B. TUNEL quantifications at 7 days post I/R. I’m confused that a-MSH was not significantly different from Sham. TUNEL results of a-MSH group have minimal variance while Shan is 0 without any variability. Can the authors include a supplementary table with the numerical data? I guess it may depend on the sample size (but the n is not specified, as mentioned before).
15. Have the authors evaluated if a-MSH improved the functionality of the retina? E.g., ERG recording to evaluate functions of outer retinal layers or behavioral test to verify that the protected RGCs are functional (e.g., optomotor response).
Minor comments
16. Redundance in lines 158-160. Note that the term “hemorrhage” is repeated in the same paragraph: “hemorrhage in the vitreous and the subretinal space...” and then, “...structural damages, hemorrhage....”.
17. Lines 248, 251, 252, 290, 291, 292. Typo. “Iab1+”, should be ‘Iba1+‘.
18. As a suggestion to the authors. Changing the color of the line for the “Untreated group” in Fig. 5C from blue to black would maintain a color code consistency with the rest of the figures.
Comments on the Quality of English LanguageEnglish is mostly well-written, Except for typos, I have not detected significant errors.
Author Response
Ng and collaborators studied the protective effects of the melanocortin alpha-melanocyte-stimulating hormone in mouse eyes afflicted with an ischemia and reperfusion model for retinal degeneration. The authors found that a-MSH prevents retinal deformation, ganglion cell loss, and delays the onset of apoptosis in affected retinas. Although a-MSH does not prevent Muller activation, it also reduces the density of microglial cells at 7 days after IR. The materials and methods are mostly described in sufficient detail, however, although the statistical analyses seem appropriate, the sample size for the different experiments is not clear. Overall, the study suggests that melanocortin may preserve cell survival in eyes with progressive degeneration caused by I/R. Although the authors have shown the efficacy of the melanocortin alpha-melanocyte-stimulating hormone in preventing neuronal loss (Brn3a and TUNEL assays), I think that glial responses would make more sense to study at earlier stages, and TUNEL signal at 24-48h do not seem to correlate with the thickness of the different retinal layers at 7d. English is mostly well-written.
Thank you for taking the time to review our manuscript. There are two issues: the immediate damage in response to the I/R and the progressive damage with time. The focus of this manuscript was to see if there were changes in the progressive damage of the retina and to start finding where, in the process, a-MSH therapy could affect. You may very well be correct in that it is the suppression of glial cell activation that preserves the eye; however, a-MSH treatment immediately protects the ganglion cells from apoptosis, suggesting more than glial cell activation but direct effects of a-MSH on the ganglion cells which do express receptors for a-MSH and others have found a-MSH to affect neuronal and RPE function See Refs 41, 53,54.
- I/R model is characterized by quick damage and degeneration. As the authors have demonstrated most of the degenerative events (TUNEL) occur within 2 days (48 hours). Then, analyzing glial reactivity (Muller cells, microglia cells, and astrocytes) would more sense at that time instead of at 7 days post I/R when degeneration is minimal. I understand that evaluation of Brn3a at 7 days confirms RGC survival, but if reactivity of glial cells is prevented by a-MSH it would be more relevant at early stages, especially when a-MSH presents a significant protection within 2 days (TUNEL assays).
The I/R model is more than quick damage it initiates a progressive degeneration of the retina. There is an immediate response, but the progressiveness of the damage also needs to be slowed. There are two potential times that a-MSH treatment could be beneficial: at the start, possibly preventing the immediate apoptotic response to I/R, and then slowing the progressive retinal damage. In addition, glial cells are not the only cells in the retina that have receptors for a-MSH. Most cells of the retina express at least one of the 4 melanocortin receptors for a-MSH. See this recent review: Wang S, et al. Therapeutic Effects of Stimulating the Melanocortin Pathway in Regulating Ocular Inflammation and Cell Death. Biomolecules. 2024 14(2):169. PMID: 38397406; PMCID: PMC10886905. The literature has shown that a-MSH can regulate non-immune cells’ survival, metabolism, and DNA repair, as well as induce anti-inflammatory factors.
- Line 123. Regarding the administration of a-MSH. Could the authors explain why the a-MSH was injected into the anterior chamber instead of Intravitreous? Would intravitreous administration improve the a-MSH efficacy?
An intravitreous injection produces a second wound site that initiates retinal damage itself. a-MSH is a 13 amino acid peptide that diffuses readily through tissues, and with an amount of a-MSH injected exceeding 10000 times the natural concentration of a-MSH at 30 pg/ml in aqueous humor) means that all parts of the eye are highly influenced by a-MSH.
- As showed by the TUNEL results, single administration delayed the apoptotic events and protected neurons. Then, why for evaluating the effects of a-MSH on the RGCs survival, Muller cells, microglia cells, and astrocytes reactivity at 7 days, the a-MSH was administrated twice (1 and 5 days after I/R). Have the authors evaluated if a single administration was also protective?
All of our anti-inflammatory assays have required two injections of a-MSH. A single injection has a transient response, whereas two have a sustained effect. This may be associated with the pharmacokinetics of using native a-MSH. We have yet to test other synthetic a-MSH analogs that have longer half-lives to see if a single injection is all that is needed.
- Lines 137-140. To facilitate reproducibility, please add the concentration of use and catalog number for the antibodies.
We provide the catalog number; the concentration will vary with the antibody production lot.
- Lines 137-140. Material and methods. I understood that Brn3a was identified by an anti-rabbit IgG ImmPRESS-AP. But I think the following sentence is confusing: “Rabbit anti-Iba1 IgG (Wako, Richmond, VA); Rabbit anti-Glutamine Synthetase IgG (Abcam, Waltham, MA) or Rabbit anti-Brn3a IgG (Abcam) with Goat anti-rabbit Cy2 (Jackson ImmunoResearch, West Grove, PA)”, and then: “For Brn3a staining, the secondary antibody was horse anti-rabbit IgG ImmPRESS-AP reagent”. Did the authors mean that Goat anti-rabbit Cy2 was used for Iba1 and Glutamine Synthetase, while the horse anti-rabbit IgG ImmPRESS-AP was for Brn3a? Can the authors clarify that?
We fixed this. There was a difference in the use of the secondary antibody. Splitting the sentence into two should now make it easier to read.
- Number of animals used in the experiment and sample size some experiments are missing. E.g., for quantifications of H&E nuclei in GCL (Fig 1D), Brn3a (Fig 2), and TUNEL (Fig. 5). For other experiments the author expressed that the sample size was n=5 (Fig. 3 legend, line 283; and Fig. 4 legend, line 325), but does it mean n=5/group?
This is listed early in all the figure legends. In Fig 1, there are eight eyes per group, and in the other figures, there are five eyes per group.
- Hemorrhages for the histology score. Can the authors explain how to determine if hemorrhages are a consequence of the elevated I/R damage or the surgical procedure? And can the authors include images of hemorrhages for readers' interpretation in Fig. 1? And have the authors investigated the hemorrhages ‘in vivo’ by examining the eye fundus with the OCT?
It is likely a response to the elevated pressure and reperfusion and not the procedure since the Sham would also have hemorrhages if it were a technical issue.
- How to determine the damage in the RPE for the histology score. E.g., the RPE in the detailed image for the “untreated group” (Fig. 1A) is damaged, but it seems to be due to the tissue processing/sectioning.
We used the standard criteria for RPE damage: the RPE cells and the layer are swollen, and the RPE may begin to round.
- Fluorescent staining in paraffin sections. Paraffin sections may not be the best for fluorescence. Cryosections would be a proper tissue processing for immunofluorescence. E.g., staining for Iba1 for the sham group (Fig. 4A) seems a bit weak and MCs seem absent in most of these layers. However, microglial cells are known to be resident microglia extended in at least 3 layers (NFL/GCL, IPL, OPL) in control conditions.
The problem is the resolution of the uploaded manuscript. We have uploaded higher-resolution images, and you can see the presence of the microglial cells (with processes) in the expected layers.
- Can the authors explain how to determine a positive microglial cell? In Fig. 4 most of the microglial cells are seen partially (the soma, or only some of their processes) ...
It is a cell body positive for Iba1. I have uploaded the higher-resolution figures to help with viewing the images.
- Line 323-324. Please double-check the following sentence because I do not observe microglial cells located in the ONL in the untreated I/R retinas: “The most noted change was the location of the Iba1+ cells in the ONL of the untreated I/R eyes versus the a-MSH-treated eyes”.
Sorry for this error; it has been corrected to OPL.
- Line 293. The suppression of the inflammatory response has not been proved in the present manuscript. Reduction of microglial cells does not prove reduced inflammatory response, because reduced microglial cells could be a consequence of the reduced RGC loss. To prove that additional experiments show a decrease in inflammatory factors, cytokines, etc.
Thank you for pointing this out, but we only suggest suppressing inflammation due to the extensive body of literature showing that a-MSH is a potent anti-inflammatory neuropeptide that also inhibits immune cell migration. This is discussed in the second paragraph of the discussion.
- Apoptosis (TUNEL) and anatomical correlation. I would expect a significant thinning of the INL and ONL at 7 days if they present a high number of apoptotic cells at 24-48 hours. However, the thickness of INL and ONL seems comparable to the sham/control group. How can those results be conciliated? Did the authors evaluate neuronal and photoreceptor loss in the INL and ONL respectively by quantifying the number of DAPI-positive rows?
Yes, it would be a reasonable assumption if it were not for the distortions of the retinal folding preventing accurate counting. Compare the ONL in the untreated images of Fig 2. The inserts also show the presence of retinal folding that results in the fusion of areas of the retina. We have tried to find a way but have been limited to the monolayers of ganglion cells and RPE, as well as staining for glial cells.
- Fig. 5B. TUNEL quantifications at 7 days post I/R. I’m confused that a-MSH was not significantly different from Sham. TUNEL results of a-MSH group have minimal variance while Shan is 0 without any variability. Can the authors include a supplementary table with the numerical data? I guess it may depend on the sample size (but the n is not specified, as mentioned before).
Thank you for pointing this out. The results are now presented with the statistics correctly labeled. The n was in the figure legend. It is 5 eyes per group.
- Have the authors evaluated if a-MSH improved the functionality of the retina? E.g., ERG recording to evaluate functions of outer retinal layers or behavioral test to verify that the protected RGCs are functional (e.g., optomotor response).
We have not done ERG with this model. We agree that if there is protection, there should be an equivalent preservation of retinal signals.
Minor comments
- Redundance in lines 158-160. Note that the term “hemorrhage” is repeated in the same paragraph: “hemorrhage in the vitreous and the subretinal space...” and then, “...structural damages, hemorrhage....”.
This has been corrected.
- Lines 248, 251, 252, 290, 291, 292. Typo. “Iab1+”, should be ‘Iba1+‘.
Corrected.
- As a suggestion to the authors. Changing the color of the line for the “Untreated group” in Fig. 5C from blue to black would maintain a color code consistency with the rest of the figures.
Corrected.
Reviewer 3 Report
Comments and Suggestions for Authors
The article "Alpha-melanocyte Stimulating Hormone Maintains Retinal Homeostasis after Ischemia/Reperfusion" is of great interest to the scientific community focusing on ocular pathologies.
The results presented in the article help to clearly define the role of MSH in maintaining healthy retinas, as well as in improving those with pathology. The presented results showed in murine models that treatment with α-MSH (alpha-melanocyte-stimulating hormone) mediated protective characteristics similar to those observed in other treated diseases. Additionally, it was demonstrated that there was protection of retinal structure, decreased loss of ganglion cells, and apoptosis. However, there was still significant activation of Müller cells and an increase in the number of macrophage/microglial cells observed. It is highly likely that macrophage/microglial cells are mediating anti-inflammatory activity and suppressing oxidative activity, which needs to be tested. One aspect to highlight is that this article leaves doors open for future research and it would be of great interest to test it in humans in the future, as it may open up future therapeutic targets.
One consideration concerns the quality of the images: they should be reviewed and enhanced for greater contrast; the introduction of symbols (arrows, asterisks, etc.) delineating different areas under study would be appreciated.
Thank you!
Author Response
One consideration concerns the quality of the images: they should be reviewed and enhanced for greater contrast; the introduction of symbols (arrows, asterisks, etc.) delineating different areas under study would be appreciated.
Thank you for reviewing our manuscript and for your support. We have uploaded higher-resolution figures to address your concerns.
Round 2
Reviewer 2 Report
Comments and Suggestions for Authors
The authors have made considerable efforts to address the reviewer's comments, resulting in significant improvements to the manuscript. However, some key concerns remain unresolved.
1. The authors focus on the protective effects of α-MSH in the progressive phase following I/R injury. However, given that the first administration of α-MSH occurred at 1 day post-I/R, and TUNEL assay results suggest ongoing degeneration up to 48 hours (2 days) post-I/R, it's pertinent to consider whether α-MSH administration could partially influence events in the early phase.
2. It would be beneficial to include representative images depicting hemorrhage and degenerative RPE to aid readers in interpreting results. Currently, the images provided in the manuscript lack these crucial details.
3. Regarding the TUNEL assays and retinal section thickness, the authors attribute discrepancies to retinal folding. While folded tissue can occur, it's essential to ensure that images are representative. If certain areas are better preserved, it raises questions about the representativeness of the images. Inconsistent or contradictory results undermine the manuscript's integrity. If new samples or images cannot be obtained, the authors should include a limitation paragraph explaining that tissue folding may hide potential neuron loss in the INL and ONL.
4. It is notable that functional or behavioral experiments to confirm the efficacy of α-MSH treatment on retinal function or animal vision are lacking. Including a limitation paragraph acknowledging this omission would enhance the manuscript's transparency.
Comments on the Quality of English LanguageI do not detect obvious grammatical errors
Author Response
We thank the reviewer for spending their time and providing more insight into our work.
- The authors focus on the protective effects of α-MSH in the progressive phase following I/R injury. However, given that the first administration of α-MSH occurred at 1 day post-I/R, and TUNEL assay results suggest ongoing degeneration up to 48 hours (2 days) post-I/R, it's pertinent to consider whether α-MSH administration could partially influence events in the early phase.
For the TUNEL assay, in Figure 5C, we injected a-MSH immediately following the I/R. Yes, we agree with you that early injection of a-MSH should influence events in the early phase. We have edited the Fig. 5 legend to clarify this and added it to the discussion.
Added to Fig 5 legend:
- C) The number of TUNEL+ cells in I/R eyes immediately treated with a-MSH were counted over 1, 4, 24, and 48 hours.
Added to the Discussion:
Lines 425- 427: Moreover, the TUNEL staining results suggest that treatment with a-MSH may provide greater protection by inhibiting or diminishing the early-phase signaling of apoptosis.
- It would be beneficial to include representative images depicting hemorrhage and degenerative RPE to aid readers in interpreting results. Currently, the images provided in the manuscript lack these crucial details.
We have added an annotated high-resolution image of a retina from an untreated I/R eye to help with our scoring criteria—Supplemental Figure 1.
- Regarding the TUNEL assays and retinal section thickness, the authors attribute discrepancies to retinal folding. While folded tissue can occur, it's essential to ensure that images are representative. If certain areas are better preserved, it raises questions about the representativeness of the images. Inconsistent or contradictory results undermine the manuscript's integrity. If new samples or images cannot be obtained, the authors should include a limitation paragraph explaining that tissue folding may hide potential neuron loss in the INL and ONL.
We have provided high-resolution images of retinas from untreated I/R eyes, where the problems of retinal folding make measuring retinal length difficult (Supplemental Figure S1).
In addition, we have added supplemental figures S2, and S3, high-resolution images of untreated and a-MSH-treated I/R retinas stained for TUNEL to show the uniformity of the signals.
- It is notable that functional or behavioral experiments to confirm the efficacy of α-MSH treatment on retinal function or animal vision are lacking. Including a limitation paragraph acknowledging this omission would enhance the manuscript's transparency.
Yes, this is an important issue, and we have added the following to the discussion.
Lines 371 -376, last sentences of the second paragraph of the discussion:
Although we did not measure photoreceptor signaling, others have shown that a-MSH-treated diabetic and I/R diabetic eyes have protected retinal functions measured by ERG [21, 25]. These results indicate that augmenting the melanocortin pathways with a-MSH treatment effectively promotes retinal cell survival and retinal structure following IRI, which may also provide protection for retinal function.